# Ca^2+^ Signaling and Src Functions in Tumor Cells

**DOI:** 10.3390/biom13121739

**Published:** 2023-12-03

**Authors:** Antonio Villalobo

**Affiliations:** Cancer and Human Molecular Genetics Area—Oto-Neurosurgery Research Group, University Hospital La Paz Research Institute (IdiPAZ), Paseo de la Castellana 261, E-28046 Madrid, Spain; antonio.villalobo@idipaz.es

**Keywords:** calcium, cancer, calmodulin, oncogene, Src tyrosine kinases, tumor cells

## Abstract

Signaling by calcium ion (Ca^2+^) plays a prominent role in cell physiology, and these mechanisms are frequently altered in tumor cells. In this review, we consider the interplay of Ca^2+^ signaling and the functions of the proto-oncogene non-receptor tyrosine kinase c-Src in tumor cells, and the viral oncogenic variant v-Src in transformed cells. Also, other members of the Src-family kinases are considered in this context. The role of Ca^2+^ in the cell is frequently mediated by Ca^2+^-binding proteins, where the Ca^2+^-sensor protein calmodulin (CaM) plays a prominent, essential role in many cellular signaling pathways. Thus, we cover the available information on the role and direct interaction of CaM with c-Src and v-Src in cancerous cells, the phosphorylation of CaM by v-Src/c-Src, and the actions of different CaM-regulated Ser/Thr-protein kinases and the CaM-dependent phosphatase calcineurin on v-Src/c-Src. Finally, we mention some clinical implications of these systems to identify mechanisms that could be targeted for the therapeutic treatment of human cancers.

## 1. Introduction

Peyton Rous [1] was a medical doctor at the Rockefeller Institute for Medical Research in New York who discovered the transmissibility of a chicken sarcoma [2] due to infection by what later was named the Rous sarcoma virus, representing the first example of a cancer produced by a viral infection. The oncogenic potential of this virus is due to the expression of the tyrosine kinase v-Src (viral Src), which is constitutively active. The discovery of this chicken oncogene was followed by the discovery of its human homologue, the cellular proto-oncogene c-Src (cellular sarcoma kinase), and other oncogenes and viruses able to produce cancer, a history that has been masterfully described in an excellent book by Gregory J. Morgan [3].

### 1.1. c-Src/v-Src

The structural organization of c-Src shows that, at its N-terminal region, it contains a domain denoted SH4 (Src homology 4), which has a sequence that is myristoylated due to its attachment to the plasma membrane. This domain is followed by a unique domain specific to each Src-family kinase (SFK) member and the SH3 (Src homology 3) domain, a proline-rich region involved in protein–protein interactions, allowing interactions with other proteins and with the SH2 (Src homology 2) domain of the protein, for maintaining c-Src in a close, inactive conformation (see Figure 1B). The SH3 domain is followed by an SH3–SH2 linker and the SH2 domain, which recognizes and binds phospho-Tyr residues in different proteins, and importantly, with the auto-inhibitory site phospho-Tyr530 at its C-terminal tail, which is phosphorylated by CSK (C-terminal Src kinase), further maintaining c-Src in its close, inactive conformation. Following the SH2 domain is the SH1 (Src homology 1) domain, which contains the catalytic tyrosine kinase site and the auto-phosphorylation residue Tyr419. The linker region between the SH2–SH1 domains also interacts with the SH3 domain, while in its close inactive locked conformation. Upon Tyr419 auto-phosphorylation and the dephosphorylation of phospho-Tyr530, most likely by receptor-like protein tyrosine phosphatase-α (as reviewed in [4]), c-Src adopts an open, fully active conformation, as shown in Figure 1C (and as reviewed in [5]). The crystal structure of human c-Src was previously determined [6,7] (see Figure 1A), and the mechanism for c-Src-mediated Ca^2+^ mobilization and the role of CaM and phospho-Tyr-CaM in the process of c-Src activation are depicted in Figure 1C,D.

The implication of c-Src in oncogenesis has been amply studied, where the overexpression and the increased activity of this kinase are common findingss in many human cancers, which exert a prominent role in increased cell proliferation, epithelial-to-mesenchymal transition (EMT), invasiveness, and metastatic dissemination, among other processes (as reviewed in [18,19,20]). The amino acid sequences of the human proto-oncogene c-Src and the Rous sarcoma virus v-Src differ in 45 amino acid residues. These changes mostly affect the N- and C-terminal regions of the proteins (see Section 2). In addition, c-Src has 10 extra residues that are missing in v-Src, particularly 7 of them at its C-terminus, including the auto-inhibitory Tyr530, what results in the constitutive activation of the latter.

### 1.2. Other Src-Family Kinases

In addition to c-Src, the family of Src-related kinases are formed by the following cellular members: Lck (lymphocyte-specific protein tyrosine kinase), Hck (hemopoietic cell kinase), Fyn (v-Fgr/v-Yes-related tyrosine kinase), Blk (B lymphocyte kinase), Lyn (Lck/Yes-related novel tyrosine kinase), Fgr (Gardner–Rasheed feline sarcoma viral oncogene homolog), Yes (Yamaguchi sarcoma viral oncogene homolog), and Yrk (Yes-related kinase). All these non-receptor tyrosine kinases play a role in multiple cellular functions, and they have prominent roles in cancer biology (as reviewed in [21,22,23,24]). The sequences of the genes in these proteins have further confirmed their similarities.

In addition, the oncogenic viral variants v-Fgr and v-Yes also play important roles in oncogenesis. The similarity of these oncogenic viral proteins was first suspected when an inhibitory monoclonal antibody raised against v-Src also inhibited the activity of v-Fgr and v-Yes [25]. Figure 2 shows that human c-Fgr presents more variations than human c-Yes, with respect to their corresponding viral variants, particularly at their N-terminus. Also, the sequences to the putative IQ-like CaM-BDs and Ca^2+^/CaM-BD of c-Src differ from those of homologous sequences in c-Fgr and c-Yes. In addition, the sequences show a few changes between the cellular and viral forms of these kinases. There are other structure-related tyrosine kinases formed by the Brk-family, comprising Brk (breast tumor kinase), Srm (Src-related kinase with no tyrosine-regulatory and myristoylation sites), Frk (Fyn-related kinase), and Bsk (β-cells Src kinase) (see review [26]).

In previous reviews, we described the connection between Src activation and functions and Ca^2+^ signaling in normal cells [26,31]. In the present article, we concentrated on studies describing the interplay between c-Src, along with other SFKs, with Ca^2+^ signaling in tumor cells, and cells transformed by the oncogenic v-Src. Also, in this review, we covered the function of Ca^2+^-binding proteins, such as calmodulin (CaM), when acting on c-Src/v-Src, as this Ca^2+^-sensor protein directly binds to and regulates these tyrosine kinases. Finally, we considered the potential action of CaM-regulated kinases on c-Src and other SFKs, as well.

## 2. Direct Interaction of CaM with c-Src and v-Src

CaM is a small Ca^2+^ sensor protein that, in humans and other mammalians, is coded by three distinct genes localized in different chromosomes that yield distinct mRNAs, differentially expressed, but are then translated into a protein with identical amino acid sequence [32]. This protein regulates more than 300 CaM-binding proteins [33,34,35], controlling a plethora of signaling pathways and many cellular functions, in normal and tumor cells (as reviewed in [36]).

### 2.1. c-Src Activation by Ca^2+^-Dependent CaM-Binding

CaM was shown to interact with c-Src. First, the isolation of c-Src from human pancreatic tumor cells was demonstrated by CaM-affinity chromatography in a Ca^2+^-dependent manner [37]. In that work, the CaM/c-Src interaction site had been predicted to be the sequence ^204^KH**Y**K**I**RK**L**DSGG**F**^214^ (residues 204–214 corresponded to the green fluorescent protein (GST)-tagged protein), located in the SH2 domain, and the authors indicated that this sequence fit a basic 1–5–10 CaM-binding motif that had the following general sequence: (RK)(RK)(RK)(FILVW)XXX(FILV)XXXX(FILW). However, the mutation of that sequence only prevented CaM binding, in part [37], suggesting that additional CaM-binding sites could exist in c-Src. A study screening compound libraries was conducted to identify compounds that bind to this CaM-binding site, thus preventing CaM binding, and it was shown that the selected compounds inhibited the in vitro proliferation of pancreatic tumor cells [38]. However, the weak therapeutic potential of these compounds was realized, as they were not specific to tumor cells and exhibited cytotoxic effects on human non-tumorigenic fibrocystic breast epithelial MCF10A cells [38].

The N-terminus of c-Src, where myristoylation and palmitoylation occurs in order to anchor the protein to the plasma membrane, was also proposed as a site where CaM binds to the kinase in a Ca^2+^-dependent mode [39] (as reviewed in [40]). Also, it was demonstrated that myristoylation contributed to the binding of Ca^2+^/CaM to v-Src, as shown using a peptide corresponding to its N-terminus [41]. The phosphorylation of the myristoylated peptide by protein kinase C (PKC) at both Ser2 and Ser11, but not at only one of these residues alone, significantly prevented the Ca^2+^/CaM binding to the peptide [41].

In a melanoma cell line, it was demonstrated that the chemical ablation of lipid rafts negatively affected the Ca^2+^ entry into the cell via a mechanism named store-operated Ca^2+^ entry (SOCE), which started after the depletion of Ca^2+^ in the lumen of the endoplasmic reticulum, and it was mediated by the stromal interaction molecule 1 (STIM1)/Orai Ca^2+^-channel system (see Figure 1D). This prevented the Ca^2+^ -mediated activation of lipid-raft-associated CaM and the activation of c-Src located in the same membrane domain, and therefore, it prevented the activation of its downstream kinase Akt (protein kinase B) signaling pathway, thus decreasing tumor growth [42]. To our knowledge, this represented the earliest report that clearly showed the c-Src/CaM interactions demonstrated by co-immunoprecipitation and the activation of c-Src by CaM in a Ca^2+^-dependent manner [42]. Both processes were prevented by a CaM antagonist and by CaM1,2,3,4, a CaM mutant with four inactive EF-hand Ca^2+^-binding sites [42]. In a later study, we demonstrated Ca^2+^/CaM-activated c-Src in vitro, in human epidermoid carcinoma and breast adenocarcinoma cells, a process that was also inhibited by a CaM antagonist [43].

Previous research described an interesting observation: A 67 kDa cytosolic tyrosine kinase, that also interacts with Ca^2+^/CaM, phosphorylated estradiol receptors lacking ligand-binding capacity, inducing their conversion to receptors able to bind estradiol. This suggested that the reverse process, when catalyzed by a phospho-Tyr-phosphatase, could be involved in the transformation to the more dangerous hormone-unresponsive form in estradiol-responsive breast cancers [44], thus making them insensitive to tamoxifen (TMX) treatment to arrest their growth. The isolated tyrosine kinase was considered to be a member of the Src family, although its molecular identity was not established. The authors also described the estrogen receptor as a 67 kDa protein, as well, which was suspiciously identical to the molecular mass of the newly identified cytosolic tyrosine kinase. The canonical human nuclear estrogen receptor alpha has 66.2 kDa [45] (as reviewed in [46]) and the beta receptors 53–59 kDa, depending on its isoforms [45,47,48] (as reviewed in [49]). Therefore, the molecular mass of the receptor alpha was close enough to the 67 kDa receptor, as described by theses authors. As the estrogen receptor [50,51] and c-Src [42,43] were both able to directly bind Ca^2+^/CaM, this could have explained the isolation of the two molecules by CaM-affinity chromatography, as carried out in [44], without the need to adduce the direct interaction between both molecules during the isolation process. Human neuroblastoma SK-N-SH cells overexpress α-synuclein, and it was demonstrated that this protein, which also forms part of the Lewy bodies in Parkinson’s disease, enhanced the interaction of Ca^2+^/CaM with c-Src, increasing its tyrosine kinase activity [52].

### 2.2. c-Src Activation by Ca^2+^-Independent CaM-Binding

Most significantly, we also demonstrated that Ca^2+^-free CaM (apo-CaM) could bind to and activate c-Src even more efficiently than the activation exerted by the Ca^2+^/CaM complex [43] (as reviewed in [26]). This suggested that the action of CaM on c-Src was more complex than first envisioned. We identified in silico the potential of two atypical IQ-like domains as potential binding sites of apo-CaM to human c-Src; the first had the sequence ^143^**IQ**AEEWYFG**K**IT**R**^155^ located in the proximal region of the SH2 domain, and the second had the sequence ^308^**LQ**EAQVM**KK**L**R**^318^ located in the proximal region of the tyrosine kinase domain [43].

Figure 3 shows the proposed Ca^2+^-dependent and Ca^2+^-independent CaM-binding sites of human c-Src. However, the proposed Ca^2+^-dependent CaM-binding site at the N-terminus [39] (as reviewed in [40]) is not shown in Figure 3, as the crystallographic structure of the N-terminus of c-Src was missing in this structure. A phylogenetic analysis demonstrated that the two proposed atypical IQ-like CaM-binding sites [43] and the Ca^2+^-dependent CaM-binding site [37] of c-Src, were fully or highly conserved in vertebrates (as reviewed in [26]). Moreover, the three CaM-binding sites were identical in human c-Src and Rous sarcoma virus v-Src (see Figure 3C).

### 2.3. Potential CaM Binding to Other SFKs

The key residues for CaM binding in the two atypical IQ-like CaM-BDs, as described in c-Src [43], should be ^143^**IQ**xxxxxxx**K**xx**R**^155^ and ^308^**LQ**xxxxx**KK**x**R**^318^ (marked in bold). As shown in Table 1, in the first sequence, these residues are identical in c-Src, Yes, Fyn and Fgr, but in the second sequence, only in c-Src and Yes. This suggested that apo-CaM-binding may not be possible in the SFK members where these key residues are missing. In contrast, the sequence of the Ca^2+^/CaM-BD in c-Src [37] appeared to be better conserved in other SFKs (see Table 1). However, as there were some changes in the homologous sequences in other SFKs, this did not guarantee that Ca^2+^/CaM-binding would occur in other SFKs in addition to c-Src, and future experimental determinations are necessary.

The alignment of the amino acid sequences of the Ca^2+^-dependent CaM-binding domain (Ca^2+^/CaM-BD) [37] and the two proposed atypical IQ-like CaM-BD [43], likely representing apo-CaM binding sites, of human c-Src (NP_938033.1) [53], with homologous sequences of human c-Yes (P07947.3) [29], Fyn (P06241.3) [55], c-Fgr (P09769.2) [27], Lyn (P07948.3) [56], Hck (P08631.5) [57], Lck (P06239.6) [58], Blk (P51451.3) [59], and Frk (EAW48241.1) [60], were obtained from the NCBI. The Brk-family kinase member Frk is included for comparison. The sequence of human Yrk is not shown, as it was not located in the NCBI. The numbers in the sequence of c-Src includes the N-terminal methionine that is removed in the mature protein.

## 3. Phosphorylation of CaM by v-Src and c-Src

Tyrosine phosphorylation of CaM by v-Src (see Figure 1C) was first described by Fukami et al. using Rous sarcoma virus (RSV)-transformed chicken fibroblasts and tumor cells, derived from rats infected by RSV, a process that was inhibited by Ca^2+^ [61,62]. The inhibition exerted by Ca^2+^ was explained because Y99 and Y138 were located at the EF-hand Ca^2+^-binding sites III and IV of CaM, respectively, and Ca^2+^ binding occluded access to these tyrosine residues. CaM was also phosphorylated in vitro by human recombinant c-Src [63], and the phosphorylation of CaM occurred in both Y99 and Y138 was directly demonstrated using Y99F- and Y138**/**F-CaM mutants presenting as a single tyrosine residue [64]. To explore the possible biological activity of phospho-Y-CaM, we prepared phosphomimetic CaM mutants by substituting one or both tyrosine residues with acidic amino acids (glutamic acid and aspartic acid), and we demonstrated that the Y/D- and Y/E-CaM mutants exerted distinct effects on the activity of some CaM target enzymes, as compared with the effect exerted by wild-type CaM [63].

Of interest, phopho-Tyr99-CaM was shown to be an activator of phosphatidylinositol 3-kinase (PI_3_K) upon binding to the SH2 domains of its regulatory 85 kDa subunit [65]. The oncogenic K-Ras4B formed a ternary complex with PI_3_K and phospho-Tyr99-CaM. This occurred upon the interaction between the distal SH2 domain of PI_3_K and the K-Ras4B/phospho-Tyr99-CaM complex, where phospho-Tyr99-CaM was in its close Ca^2+^-free conformation; while free phospho-Tyr99-CaM, in its extended Ca^2+^-bound form, interacted with the proximal SH2 domain of PI_3_K (as reviewed in [10,66]). The activation of the PI_3_K/Akt/mTOR (mammalian target of rapamycin) pathway by this CaM-mediated mechanism played an essential role in the proliferation and survival of K-Ras-driven tumors (as reviewed in [66]).

The oncogenic protein tyrosine phosphatase receptor type Z1 (PTPRZ1), found in highly malignant small-cell lung carcinoma, induced the inactivation of this phosphatase upon binding its ligand pleiotrophin, resulting in increased levels of phospho-Tyr-CaM [67]. This suggested that CaM phosphorylated by c-Src, among other tyrosine kinases, could increase in these PTPRZ1-overexpressed tumors, contributing to its malignancy.

## 4. The Role of CaM-Dependent Kinases in SFKs

### 4.1. CaM-Dependent Kinase II

CaMK-II has been one of the best-studied CaM-regulated kinases involved in multiple cellular functions via its role in a variety of signaling pathways (as reviewed in [68,69,70]). Early reports demonstrated that v-Src, but not c-Src, was associated with a Ca^2+^/CaM-dependent Ser/Thr-protein kinase in a rat sarcoma cell line and in rat fibroblasts transformed with the Rous sarcoma virus [71,72]. At the time, this kinase was considered to be the 60 kDa subunit of the CaM-dependent protein kinase II (CaMK-II) that was able to phosphorylate a single 52 kDa endogenous substrate, and to the best of our knowledge, its nature has yet to be fully understood [72]. The phosphorylation of c-Src by several Ser/Thr-protein kinases was demonstrated, but CaMK-II was not reported among them (as reviewed in [73]). Therefore, we used the phosphorylation prediction NetPhos 3.1 server from DTU Health Tech (available from https://services.healthtech.dtu.dk/services/NetPhos-3.1/, accessed 28 November 2023) to predict the potential CaMK-II-mediated phosphorylation site(s) in c-Src and v-Src. We found that the phospho-Ser/Thr prediction scores by c-Src/v-Src, ranged between 0.401 and 0.479, values which were, in some cases, slightly above the score of Thr101 in c-Src or below the set threshold. However, in human c-Src, the phosphorylation motifs of different CaM-dependent kinases, including CaMK-IIδ [74], of general sequence -(R/K)-X-X-p(S/T)-X-(D/E) [75], where X indicates any amino acid residue, coincided with the sequence ^40^KPA**S**AD^45^ [53], and this suggested that the Ser43 in the mature protein could be phosphorylated by CaMK-IIδ (see Figure 4A).

The phosphorylation of the Src-family kinase Lck (lymphocyte-specific protein tyrosine kinase) by CaMK-II was inferred in human leukemic Jurkat T cells by artificially increasing intracellular Ca^2+^ and using an inhibitor of CaMK-II [84] (see Figure 4). This Ca^2+^ increase, followed by CaM-mediated CaMK-II activation, triggered the activation of the extracellular regulated kinases 1 and 2 (ERK1/2) [85], as well as the activation of apoptosis via the phosphorylation of the adaptor protein p66Shc [86].

Chloride mobilization appears to control CaMK-II and c-Src activities in breast adenocarcinoma cells, as chemically inhibiting or knocking down the chloride-channel ANO1 (anoctamin-1) with a shRNA (short hairpin RNA), reduced both epidermal growth factor receptor (EGFR) and CaMK-II activation, inducing afterwards a decrease of Akt/c-Src/MAPK (mitogen-activated protein kinase) activation, both in vitro cultured cells and in vivo using xenographted tumor cells, inhibiting therefore cancer progression, and underscoring the tumor promoting role of these pathways [87]. Moreover, stimulating pancreatic tumor cells with UTP or suramin, an agonist of the nucleotide receptor P2Y2, activated cell proliferation by a pathway implicating CaMK-II and c-Src, likely using mechanisms similar to the ones described above for the CaMK-II/c-Src/MAPK pathway in breast cancer cells [88].

### 4.2. Death-Associated Protein Kinase

DAPK (death-associated protein kinase) is another CaM-regulated Ser/Thr kinase that exerts a tumor-suppressing function by inhibiting cell proliferation and cell adhesion and, most relevantly, inducing apoptosis (as reviewed in [89,90]). c-Src inhibited DAPK by phosphorylating its residues Tyr491 and Tyr492, a process that was enhanced in tumor cells with aberrantly elevated c-Src activity [91]. The receptor tyrosine phosphatase, denoted LAR (leukocyte common antigen-related) (as reviewed in [92]), dephosphorylated Tyr491/ Tyr492 (see Figure 4B). However, activating the EGFR pathway in the tumor cells activated c-Src, which subsequently downregulated LAR, suggesting this process could be synergistic with DAPK inactivation in order to evade apoptotic cell death [91].

### 4.3. Eukaryotic Elongation Factor-2 Kinase

The eukaryotic elongation factor-2 kinase (eEF-2K), previously known as CaMK-III, is a Ca^2+^/CaM-dependent kinase, belonging to the atypical α-kinase family that controls protein synthesis, and functions as a major regulator of autophagy by acting downstream of mTOR. eEF-2K activity was also synchronized with the different phases of the cell cycle and was overexpressed in many cancerous cells promoting cell survival, enhanced proliferation, and metastatic dissemination, under hypoxic and nutrient-deprived conditions (as reviewed in [93,94]). Similarly, using human ductal pancreatic PaCa tumor cells, it was shown that eEF-2K controlled the activity of c-Src via tissular transaminase-2, inducing cell proliferation, increased motility, extracellular proteolysis by matrix metallopeptidase-2 (MMP-2), and, thus, the invasion and dissemination of tumor cells (see Figure 4C). In general, this takes place via the β1-integrin/Src/urokinase plasminogen activator surface receptor (uPAR)/MMP-2 pathway [95].

## 5. The Role of the CaM-Dependent Phosphatase Calcineurin in SFKs

*N*-myristoyltransferase (NMT) is the enzyme implicated in the myristoylation of SFKs, and it was shown that Lyn (Lck/Yes-related novel tyrosine kinase) and Fyn (v-Fgr/v-Yes-related tyrosine kinase), and to a lesser extent, Lck, phosphorylated the N-terminus of NMT, and its phosphorylated form was then dephosphorylated by the Ca^2+^/CaM-dependent phosphatase calcineurin (CaN), which was also denoted as protein phosphatase 2B (PP2B) [96] (as reviewed in [97]). While this may initially appear anomalous, as CaN has generally been considered a Ser/Thr-phosphatase (as reviewed in [98]), the role of CaN dephosphorylating phospho-tyrosine residues has been documented in other instances. For example, CaN dephosphorylated the transcription factor STAT3 (signal transducer and activator of transcription 3) when it had been phosphorylated by c-Src and where the possibility of the involvement of the dual Ser/Thr-Tyr-phosphatase MKP-1 (mitogen-activated protein kinase phosphatase-1) had been excluded [99]. In addition, an old report had already demonstrated that CaN dephosphorylated EGFR-(Tyr)-phosphorylated substrates [100], further demonstrating that CaN was indeed a dual Ser/Thr-Tyr phosphatase.

## 6. Ca^2+^ and v-Src Functions in Transformed Cells

### 6.1. v-Src as Modulator of Ca^2+^ Channels

The oncogenic v-Src tyrosine kinase has been shown to exert important Ca^2+^-mediated cellular functions. To experimentally demonstrate this, v-Src-transformed cells were used. To this end, a temperature-sensitive v-Src mutant was transfected in rat pheochromocytoma cells, and it was shown that it increased the expression of diverse Ca^2+^ channels at the permissive temperature, similar to the results after differentiation was induced by neural growth factor (NGF) [101,102]. Particularly, specific Ca^2+^-channel blockers for different channel types were identified at the permissive temperature (37 °C), affecting the expression of a significant percentage (~50%) of N-type channels, a small percentage (~8%) of L-types channels, and a significant proportion (~42%) of channels that were insensitive to inhibitors specific for the N- and L-types. This contrasted with the percentages found at the non-permissive temperature (40 °C), ~30%, ~46%, and ~24%, respectively [101]. The changes in the expression of the Ca^2+^ channels may contribute to changes in Ca^2+^ fluxes, as observed in v-Src-transfected rat fibroblasts [103].

In addition, v-Src-transformed mouse fibroblasts presented an additional electrical current due to Ca^2+^-activated K^+^ channels, either because of de novo expression or the activation of previously silent channels [104], an important point that was not solved in this report.

### 6.2. v-Src and Cell Proliferation

The ability of tumor cells to proliferate in a Ca^2+^-deprived medium appeared to be mediated by v-Src. This was demonstrated in rat fibroblasts transformed with the temperature-sensitive v-Src mutant, as previously mentioned, and the cells did not proliferate at the non-permissive temperature, where the v-Src expression had been prevented, but rapidly initiated DNA synthesis and proliferation after transferring the cells to the permissive temperature, which allowed for v-Src expression [105,106]. The authors of the report pinpointed a target affecting the G_1_/S transition of the cell cycle, where the oncogenic transfected kinase had exerted its action without affecting the role of Ca^2+^/CaM at this step of the cell cycle [105,106]. It was suggested that the carcinogenic potential of the sarco/endoplasmic reticulum Ca^2+^-ATPase (SERCA) inhibitor thapsigargin was related to the increase in the cytosolic Ca^2+^ concentration, as it had prevented the refilling of the endoplasmic reticulum, leading to the rise in the cytosolic Ca^2+^ concentration inducing the subsequent Src-mediated activation of the MAPK pathway, which then triggered the proliferation in the cells transfected with v-Src [107].

### 6.3. v-Src and Ca^2+^-Signaling Effectors

Also, v-Src expression modified cellular Ca^2+^ signaling. The transformation of rat fibroblasts with v-Src strongly amplified the endothelin-induced production of inositol-(1,4,5)-trisphosphate (IP_3_) and the transient increase in cytosolic Ca^2+^ concentration [108]. On the contrary, the v-Src expression in these transformed fibroblasts decreased the thrombin- and lysophosphatidic acid (LPA)-mediated IP_3_ production and the transient cytosolic Ca^2+^ peaks induced by these effectors [108,109].

### 6.4. v-Src and MARCKS Function

In v-Src-transformed NIH3T3 murine fibroblasts, a decreased expression and an enhanced translocation of the Ca^2+^/CaM-regulated enzyme MARCKS (myristoylated alanine-rich C-kinase substrate) from the plasma membrane to the cytosol was demonstrated, indicating that v-Src had activated PKC [110]. The authors suggested that the changes in the MARCKS location could have been responsible, at least in part, for the altered cytoskeletal organization and the changes in morphology, as observed in the transformed cells.

### 6.5. v-Src and Gap-Junction Communication

In v-Src-transformed fibroblasts, it was shown that this kinase phosphorylated connexin43, disrupting gap-junction communication, a functional process shared with the unrelated Fujiyama sarcoma virus tyrosine kinase p130gag-fps [111]. This should block the cell-to-cell transfer of calcium ions and the Ca^2+^-mobilizing signaling molecule IP_3_, a phenomenon demonstrated in the human cervix adenocarcinoma HeLa cells transfected with different connexins, including connexin43 [112].

## 7. Ca^2+^ and SFKs Functions in Tumor Cells

In this section, we considered some examples of the involvement of Ca^2+^ in c-Src and other SFKs functions, in tumor cells. 

### 7.1. Role of Ca^2+^ on c-Src Degradation

The overexposure of the estrogen-dependent human breast cancer cell line T-47D to exogenous Ca^2+^ activated the Ca^2+^-dependent protease calpain, which proteolyzed c-Src. This resulted in decreased c-Src signaling via the PI_3_K/Akt survival pathway, inhibiting tumor cell proliferation in vitro and decreasing tumor growth in vivo when the tumor cells were xenografted in BALB/c nude mice [113]. Moreover, the exogenous Ca^2+^-mediated c-Src degradation in human triple-negative breast cancer MDA-MB-231 cells resulted in the inhibition of c-Src-mediated EGFR transactivation, contributing to decreased proliferation [114].

### 7.2. Role of Ca^2+^ and c-Src in Tumor Cells Invasiveness

The development of a fluorescent Ca^2+^ biosensor tethered to the N-terminus of the Orai channels allowed the direct detection of the subplasmalemmal Ca^2+^ increase in a single invadopodium of human melanoma cells due to the activation of SOCE, which resulted in the activation of the Ca^2+^/CaM-dependent proline-rich tyrosine kinase 2 (Pyk2) [115] plus the c-Src signaling pathway [116], which was essential for starting the invasiveness of these malignant tumors. In an apparently conflicting report using STIM1-knockout prostate cancer cells, it was proposed that Ca^2+^ entry via SOCE was not required for the activation of CaMK-II and the c-Src/MAPK pathway, and it instead appeared that STIM1 was a downstream target of the c-Src/MAPK pathway because ERK1/2 had phosphorylated STIM1 [117]. Nevertheless, this does not nullify the need of Ca^2+^ for the activation of the c-Src/CaMK-II/MAPK pathways, as enough cytosolic Ca^2+^ could be derived from intracellular stores.

c-Src participated in the assembly and disassembly of tight junctions, functions that were mediated by the signaling pathways involving calcium ions [118,119]. Also, Ca^2+^-dependent, integrin-mediated cell adhesion to the extracellular matrix and cadherin-based cell–cell contacts were disassembled during the epithelial–mesenchymal transition (EMT), where were required as early steps in order to enable the invasion of adjacent tissues and the migration of the tumor cells to distant organs, resulting in metastases.

The disassembly of cell–cell contacts in carcinomas was concurrent with increased cell survival and anchor-independent growth, and c-Src, which has enhanced activity in many carcinomas, participated in these processes. Thus, in a non-invasive mouse breast carcinoma cell line transfected with a constitutive active chicken c-Src mutant, a continual phosphorylation of Src at Tyr418 was shown, but it required Ca^2+^-dependent integrin-mediated cell adhesion to phosphorylate the focal adhesion kinase (FAK) at Tyr577, as well as the interaction of c-Src with phospho-FAK. Therefore, the disruption of the Ca^2+^-dependent cell adhesion during the anchor-independent growth had induced the dissociation of the Src/FAK complex and the dephosphorylation of FAK [120].

The metalloprotease ADAM10 (disintegrin and metalloproteinase domain-containing protein 10) is regulated by c-Src and CaM [121,122,123]. ADAM10 was overexpressed in high-grade pituitary adenoma cells, and the invasiveness of these tumors was explained, at least in part, by the c-Src/CaM-mediated mechanisms [121,122]. The inactive form of this proteinase, denoted pro-ADAM10, was activated upon Ca^2+^ influx by reducing the Ca^2+^/CaM-pro-ADAM10 interaction, and the activated proteinase subsequently induced the cleavage of the adhesion molecules CD44 and L1CAM (cell adhesion molecule L1) [123], disrupting the cell–cell interaction, detaching cells from the extracellular matrix, and thus enhancing the invasiveness of the tumor cells. In addition to this CaM-mediated mechanism, it was demonstrated that the activation of PKC with phorbol ester promoted c-Src activation, and the active kinase was shown to interact with ADAM10, where c-Src appeared to compete with CaM for ADAM10 binding [121,122], further promoting the activation of the proteinase, inducing the cleavage of adhesion molecules, and increasing the invasiveness of the tumor cells.

### 7.3. Role of Ca^2+^ and c-Src in Cell Migration and Tumor Progression

Adducin is an oligomeric (mostly dimers and tetramers) CaM-binding protein that is involved in recruiting, bundling, and capping the fast-growing end of actin filaments, as well as spectrin recruitment, among other functions. Both Ca^2+^/CaM-binding and PKC phosphorylation inhibited adducin functions by controlling membrane ruffling and myosin-based cell motility (as reviewed in [124]). As the expression of adducin isoforms is altered in tumorigenesis, the role of α-adducin and γ-adducin was studied in the migration and invasiveness of cells from non-small-cell lung cancer (NSCLC). It was determined that α-adducin had been delocalized and γ-adducin had been highly downregulated, and these had been concomitant with the development of an invasive mesenchymal phenotype. These activities were reverted by the forced overexpression of α-adducin, but not γ-adducin, which was concomitant with the increased assembly of the focal adhesions and the hyperphosphorylation of c-Src and paxillin [125].

Curiously, increased levels of bile deoxycholic acid (DCA) were correlated to the development of colorectal adenomas and the progression to carcinoma. In this context, in human DCA-treated colorectal adenocarcinoma HT-29 cells, increased intracellular Ca^2+^ and CaMK-II activation was shown, which promoted signaling via the EGFR-mediated MAPK activation upon c-Src recruitment and the phosphorylation of the receptor at Tyr845 [126]. As discussed by the authors, these findings could explain, at least in part, that excessive DCA production may promote colorectal tumor growth.

### 7.4. AKAP12 and Ca^2+^/c-Src Signaling

Some scaffolding proteins have the ability to negatively regulate tumor progression and metastases. Among them, AKAP12 (A-kinase anchoring protein 12), also known as gravin, has one binding site for c-Src and four binding sites for Ca^2+^/CaM, belonging to the class denoted 1-5-10 motifs (as reviewed in [127,128]), among many other binding sites for different proteins. The suppression of metastasis by AKAP12 was mediated, at least in part, by inhibiting the Src-regulated angiogenesis of distal blood vessels mediated by VEGF (vascular endothelial growth factor) (as reviewed in [128]). Less is known about the effect of CaM on AKAP12, although the phosphorylation of the scaffold protein by PKC prevented CaM binding (as reviewed in [127]). This competition was first described in SSeCKS (Src-suppressed C kinase substrate), the rodent orthologue of human AKAP12 [129]. It was suggested that the role of CaM binding to SSeCKS/AKAP12 proteins could most likely be to prevent the activation of Ca^2+^/CaM-dependent kinases by sequestering CaM [129].

### 7.5. Wnt/Ca^2+^ Signaling and c-Yes Function

The different Wnt (Wingless: integration protein-1) signaling pathways regulate many cellular functions and play important roles in the development of metastasis, for example, in bones (as reviewed in [130,131]). Upon the Wnt ligand binding to its plasma membrane-bound receptor Frizzled and other co-receptors, one of the pathways activated was the non-canonical Wnt-5a/Ca^2+^ pathway, which activated the transcription factor NFAT1 (nuclear factor of activated T cells 1) after its dephosphorylation by Ca^2+^/CaM-activated calcineurin and its translocation from the cytosol to the nucleus. Using human mammary epithelial cells and MCF7 adenocarcinoma cells, it was shown that the NFAT1 activation by the Wnt-5a/Ca^2+^ pathway was suppressed via a pathway involving Wnt-5a, the SFK c-Yes, Cdc42 (clone derived 42), and CK1α (casein kinase 1α), where c-Yes, but not c-Src or c-Fyn, had been activated by Wnt-5a [132]. Therefore, the authors proposed that the loss of Wnt-5a expression and reduced c-Yes/Cdc42/ CK1α signaling may increase the metastatic potential of breast cancer through the hyperactivation of NFAT.

### 7.6. SFKs and Ca^2+^ Signaling in the Immune Response

The activation of the CD3 (clone derivate 3)/TCR (T-cell receptor) complex with an immobilized anti-CD3 antibody in leukemia Jurkat cells increased the cytosolic Ca^2+^ and induced PKC activation upon translocation to the plasma membrane. In turn, PKC phosphorylated c-Src at Ser12, resulting in interleukin-2 production and the induction of cell proliferation [133]. However, using a soluble anti-CD3 antibody did not induce interleukin-2 production but, instead, blocked cell proliferation. This was in contrast to the behavior of normal T cells, where both soluble and immobilized anti-CD3 antibodies induced interleukin-2 production and stimulated cell proliferation [133]. The transmembrane phospho-Tyr-phosphatase CD45 (clone derived 45) also participated in TCR signaling as well as the dephosphorylation of the C-terminal negative regulatory Tyr residue of SFKs (as reviewed in [134]). The dephosphorylation of this tyrosine set the kinase in its open, active conformation [73]. In a report using a Jurkat cell line depleted of CD45 and transfected with different EGFR/Lck-fusion chimera proteins, it was demonstrated that Ca^2+^ signaling by TCR only occurred by the activated EGFR/Lck chimera after CD3 activation and that an inactive form of EGFR/Lck failed to induce Ca^2+^ signaling [135].

It was demonstrated that upon the activation of BCR (B-cell receptor) when binding antigens or anti-receptor antibodies, the active receptor induced a Ca^2+^ release from the intracellular stores, which was followed by the extracellular Ca^2+^ entry via the Ca^2+^ release-activated ion channels, and some tyrosine kinases intervened in this process (as reviewed in [136]). In the cell line DT40, a chicken B-cell lymphoma derived from the bursa of Fabricius, it was reported that Lyn and the unrelated non-SFK tyrosine kinase Syk (Syk family) were absolutely necessary for activation of Ca^2+^-release-activated Ca^2+^ channels (CRAC) after emptying the intracellular Ca^2+^ stores, and this effect was not due to the increased expression of the Orai1 channel or the STIM1 Ca^2+^ sensor, but to unknown phosphorylation events mediated by these tyrosine kinases in order to maintain the function of the involved target proteins [137]. Also, using wild-type and different mutants of DT40 cells subjected to a massive Ca^2+^ load upon treatment with the ionophore ionomycin to bypass BCR Ca^2+^ signaling, the participation of Syk and Lyn, but not Fyn or Lck, was established in BCR-independent Ca^2+^-induced apoptosis [138].

## 8. Medical Implications

In this section, we describe findings relating the activation or the inhibition of some SFKs with mechanisms involving Ca^2+^-dependent events that could be of medical interest in the field of oncology. Table 2 summarizes some of these processes, and then we illustrate these events in more detail.

The table contains information obtained primarily from human tumor cells, with a few examples from animal tumor models. Please note the following: 5-HT_6_, serotonin receptor-6; Bcr-Abl, break point cluster region-Abelson tyrosine kinase; Ca^2+^/CaM-BD, Ca^2+^/calmodulin-binding domain; CD20, clone derived 20; cisplatin, *cis*-dichlorodiammine platinum (II); CML, chronic myeloid leukemia; doxorubicin, (7*S*,9*S*)-7-[(2*R*,4*S*,5*S*,6*S*)-4-amino-5-hydroxy-6-methyloxan-2-yl]oxy-6,9,11-trihydroxy-9-(2-hydroxyacetyl)-4-methoxy-8,10-dihydro-7*H*-tetracene-5,12-dione; DMBA, 7,12-dimethylbenz(*a*)anthracene; eFF-2K, eukaryote elongation factor-2 kinase; MPNST, malignant peripheral nerve sheath tumor; ND, not described; PI_3_K, phosphatidylinositol 3-kinase; PLCγ, phospholipase-Cγ; PP1, 4-amino-5-(4-methylphenyl)-7-(t-butyl)pyrazolo-d-3,4-pyrimidine; SFK, Src-family kinase; TFP, trifluoperazine; TMX, tamoxifen.

Although little progress has been made in regards to inhibiting Src and other SFKs in order to treat cancer due to the complexity of the signaling pathways involved in the control of many cellular functions and the development of resistance, increased efforts open new hopes for success, particularly using combinatorial therapeutic regimes (as reviewed in [5,154,155]). Among the strategies used to develop SFK inhibitors, one of particular interest used functionally silent active-site mutants to sensitize the target kinase against an inhibitor that did not affect the wild-type kinases. Using this technique, N^4^-derivatized and C(3)-derivatized analogs of PP1 (a synthetic pyrazolo [3,4-d]pyrimidine) were developed with increased capacities to inhibit v-Src, c-Src mutants, and the SFKs Lck and c-Fyn (as reviewed in [139]). Also, as previously mentioned, targeting the proposed Ca^2+^/CaM-binding site of c-Src located at the SH2 domain with different chemical compounds [37] was initiated as a potential therapeutic strategy to inhibit pancreatic tumor growth [38]. Nevertheless, in a cell line of normal breast cells, the cytotoxic effects of most of these compounds were also identified in this study, suggesting that this strategy may not be too promising.

The monoclonal antibody rituximab against the CD20 antigen expressed in neoplastic B cells was used in a clinic to treat non-Hodgkin lymphomas, usually in combination with chemotherapy [156]. Rituximab induced the crosslinking of lipid-raft-associated CD20 and the apoptosis of the tumor cells, and these processes were shown to be mediated by the activation of certain Src-family kinases (SFKs), which induced intracellular Ca^2+^ mobilization, as shown in Ramos B cells [147,148], a cell line derived from a Burkitt’s lymphoma of a three-year-old patient. The oncogenic fusion protein Bcr-Abl (breakpoint cluster region–Abelson tyrosine kinase) was expressed in human chronic myeloid leukemia cells (as reviewed in [157]), and the effective treatment of these cells with the tyrosine kinase Imatinib started to fail when mutations appeared in the Bcr-Abl protein [142]. Therefore, additional strategies have been considered to treat Imatinib-resistant patients. In this context, the combined inhibition of c-Src and PI_3_K, which blocked the PI_3_K/Akt/mTOR survival pathway, induced in the presence of Imatinib apoptosis and autophagy in Bcr-Abl-positive leukemia cells. These processes are due to the inhibition of SERCA and Ca^2+^ released via IP3 receptors [142]. Curiously, the dual c-Src and Bcr-Abl second-generation kinase inhibitor bosutinib, for the treatment of chronic myeloid leukemia, was shown to inhibit other kinases, and most relevantly, CaMK-IIγ by blocking its ATP-binding site, a mechanism that was distinct of other known CaMK-II inhibitors [143].

As in the case of c-Src, the CaM-dependent kinase Pyk2 contributed to the transactivation of EGFR, increasing cell survival and proliferation while favoring invasiveness and metastasis development. In a study performed with glioblastoma cell lines, it was shown that the tyrosine kinase inhibitor tyrphostin A9 (also known as malonoben), in addition to inhibiting EGFR and other tyrosine kinase receptors, also inhibited Pyk2; reduced glioblastoma cell proliferation and migration; and induced cell death by apoptosis [158]. Also, the experimental c-Src inhibitor sacaratinib, in combination with the CaM antagonist trifluoperazine (TFP), enhanced the inhibitory capacity of other inhibitors of the ErbB3 signaling pathway, and this was demonstrated by blocking the proliferation and survival of malignant human Schwannoma cells [146].

The treatment of cancer with cisplatin induced the expression and the activation of the SFK Lyn in macrophages, a process that was regulated by Ca^2+^/CaM and CaM-dependent kinases [149]. Also of medical interest is the observation that in addition to the well-known cardiotoxic effects of the widely used anti-cancer drug doxorubicin (as reviewed in [144]), this drug also exerted ovarian toxicity. And, in this case, its action was mediated by an abnormal increase in the cytosolic Ca^2+^ in the follicular cells because of its massive release from the endoplasmic reticulum, which had been mediated by c-Src activation [145].

The monoclonal antibody TRA-8 against the death receptor-5 (DR5) acted as an agonist, inducing apoptosis in pancreatic tumor cells. However, resistance usually developed, and additional therapeutic strategies have been implemented in order to solve this problem. Of interest to our topic in this review, TFP and TMX were used as adjuvants to overcome this resistance, and c-Src was implicated in its mechanism of action. Thus, in TRA-8-resistant pancreatic tumor PANC-1 cells, TFP or TMX alone did not induce apoptosis but increased TRA-8-induced apoptosis in a dose-dependent manner [140]. The reason for this apparently contradictory observation was that the death-inducing-signaling complex (DISC) associated to DR5 activation by TRA-8 recruited c-Src/CaM, which elicited survival signals. Therefore, the inhibition of CaM by TFP/TMX had prevented c-Src/CaM-mediated survival signaling. But also, TFP/TMX targets the transcription factor Sp1 (specificity protein-1) acting on the DR5 gene increasing the expression of this receptor, allowing both processes efficient apoptotic cell death of the tumor cells [140].

A study explored the viability of a gene therapy approach to treat the highly malignant triple-negative breast cancer (TNBC), which lacks the expression of ErbB2/HER2, estrogen, and progesterone receptors, by targeting the Ca^2+^/CaM-dependent kinase eEF-2K, which was highly expressed in these tumors [141]. The administration of nanoparticles containing the tumor-suppressor micro-RNA miR-603 to TNBC-cell-xenografted mice inhibited the expression of eEF-2K and the activity of downstream signaling systems, including c-Src, inhibiting tumor cell growth, cell migration, and invasiveness. This was due to the fact that in untreated cancerous cells, eEF-2K has a prominent role promoting tumorigenesis and the associated malignant features [141].

Some carcinogenic polycyclic hydrocarbons have been known to activate Fyn and Lck in the human leukemia T-cell line HPB-ALL, inducing tyrosine phosphorylation and the activation of phospholipase-Cγ, the synthesis of IP_3_, and the subsequent release of Ca^2+^ from the endoplasmic reticulum, thereby increasing cytosolic Ca^2+^ levels [151]. In contrast, in the mouse-thymus-derived lymphoma WEHI 7.1 cell line, low concentrations of the glucocorticoid dexamethasone were shown to transform the TCR-induced transient cytosolic Ca^2+^ increase to an oscillatory behavior upon strong receptor stimulation, and this was due to the Lck inhibition, which bound to and positively regulated IP_3_ receptors type I, while upon weak TCR stimulation, the Ca^2+^ oscillations stopped [152]. As discussed by the authors of this report, Lck-mediated mechanisms may explain the suppression of the immune response mediated by glucocorticoids.

In stably transfected human astrocytoma 1321N1 cells, it was demonstrated that, as expected, a newly developed agonist of the serotonin (5-hydroxytryptamine) receptor 5-HT_6_ induced a cytosolic Ca^2+^ surge, followed by the activation of both Fyn and its downstream MAPK pathway [153]. The degranulation of P815 mast cells, isolated from a mouse mastocytoma and induced by an allergenic compound, caused Lyn activation and Ca^2+^ mobilization via the opening of the IP_3_ receptor Ca^2+^ channels [150].

## 9. Concluding Remarks

The major issues remaining in the areas covered by this review are as follows: (i) To determine which is/are the functional action(s), if any, of the direct interaction of Ca^2+^/CaM, and possibly apo-CaM, on the enzymatic activity of v-Src/c-Src, by potentially exerting regulatory functions on these kinases; (ii) To determine whether phospho-Tyr-CaM exerts any role on v-Src/c-Src functions; and (iii) To determine whether CaM binds and regulates other SFKs.

Also, it would be interesting to further investigate whether CaMK-II, and other CaM-dependent kinases, are capable of phosphorylating c-Src/v-Src and other SFKs, despite the negative analysis that we obtained using the phosphorylation prediction tool of the NetPhos 3.1 server (see Section 4). Particularly, when some positive experimental results had been obtained in human Lck using tumoral Jurkat T cells [84], the analysis that we then performed using the same tool to search for putative phospho-Ser/Thr sites in Lck phosphorylated by CaMK-II resulted in very low prediction scores (407–469). Nevertheless, the authors could not conclude that CaMK-IV, instead of CaMK-II, was the kinase phosphorylating Lck [84]. Nevertheless, it is well known that informatics-based prediction tools are not always accurate methods of analysis, and only experimental demonstrations can solve these issues.

We expect that additional studies on the Ca^2+^ signaling mechanisms and CaM-regulated systems, affecting the functions of oncogenic non-receptor tyrosine kinases belonging to the Src family, may identify new therapeutic targets that can block the hyperactivity and/or overexpression of these kinases in tumor cells with specificity, in order to treat patients with SFK-driven cancers.

## Figures and Tables

**Figure 1 biomolecules-13-01739-f001:**
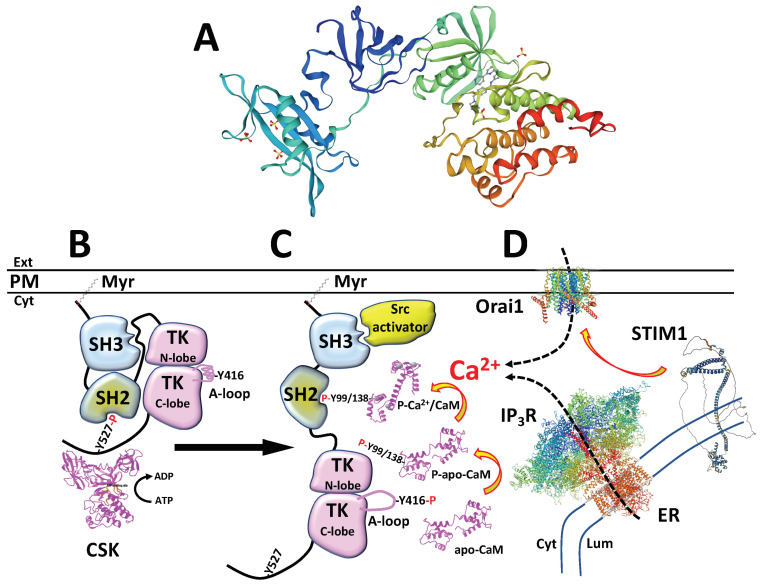
c-Src activation and Ca^2+^/CaM signaling**.** (**A**) The figure shows the crystal structure derived from X-ray diffraction at 1.91 Å resolution of unphosphorylated human c-Src in complex with a desmethyl analog of the antileukemia drug Imatinib [8]. The structure (SMTL ID: 1y57.1), depicted in a rainbow color code going from the N-terminus (blue) to the C-terminus (red), was built using the UniProt SWISS-MODEL database from Biozentrum, University of Basel, feeding the system with the amino acid sequences obtained from the NCBI (National Center for Biotechnology Information). Five sulfate ions not bound to c-Src are also shown. (**B**) The drawing (adapted in part from [7]) shows a model of the close, inactive conformation of c-Src, where phospho-Tyr527, phosphorylated by C-terminal Src kinase (*CSK*), binds to the SH2 domain, facilitating the interaction between the SH2/SH3 domains and the interaction of the linker SH2/N-lobe of the tyrosine kinase domain (*TK N-lobe*) with the SH3 domain. (**C**) The drawing also shows the open, active conformation of c-Src upon dephosphorylation of Tyr527 and interaction of the SH3 domain with Src-activators containing proline-rich domains [9], exposing the activation loop (*A-loop*) with auto-phosphorylated Tyr416. Active c-Src is able to phosphorylate apo-calmodulin at Tyr99/138 (*P-apo-CaM*) [10], which in turn binds to the SH2 domain (not shown), and also upon Ca^2+^-binding to phospho-CaM (*P-Ca^2+^/CaM*). (**D**) c-Src-mediated Ca^2+^ mobilization and formation of the Ca^2+^/CaM complex occurs upon cytosolic Ca^2+^ increase after its release from the endoplasmic reticulum (*ER*) via the inositol-1,4,5-trisphosphate receptor (*IP_3_R*), followed by Ca^2+^ entry from the extracellular medium via the Ca^2+^-channel Orai1 upon depletion of the ER by the Ca^2+^ sensor stromal interaction molecule 1 (*STIM1*), which interacts with Orai1 upon by changing its conformation and extending its distal segment. For more details, see text. The structures of *Xenopus laevis* apo-CaM (NCBI ID: 1CFD) [11], identical to human CaM; human Ca^2+^/CaM (NCBI ID: 1CLL) [12]; human IP_3_R1 (UniProt ID: Q14643) [13]; *Drosophila melanogaster* Orai1 (UniProt ID: Q96D31), with 73% identity to human Orai1 [14]; rat CSK (NCBI ID: 1K9A) [15]; and STIM1 (UniProt ID: AF-Q13586-F1) [16,17] were taken from NCBI and UniProt SWISS-MODEL databases. Cyt, cytosol; Ext, extracellular medium; Lum, endoplasmic reticulum lumen; Myr, myristic acid; PM, plasma membrane.

**Figure 2 biomolecules-13-01739-f002:**
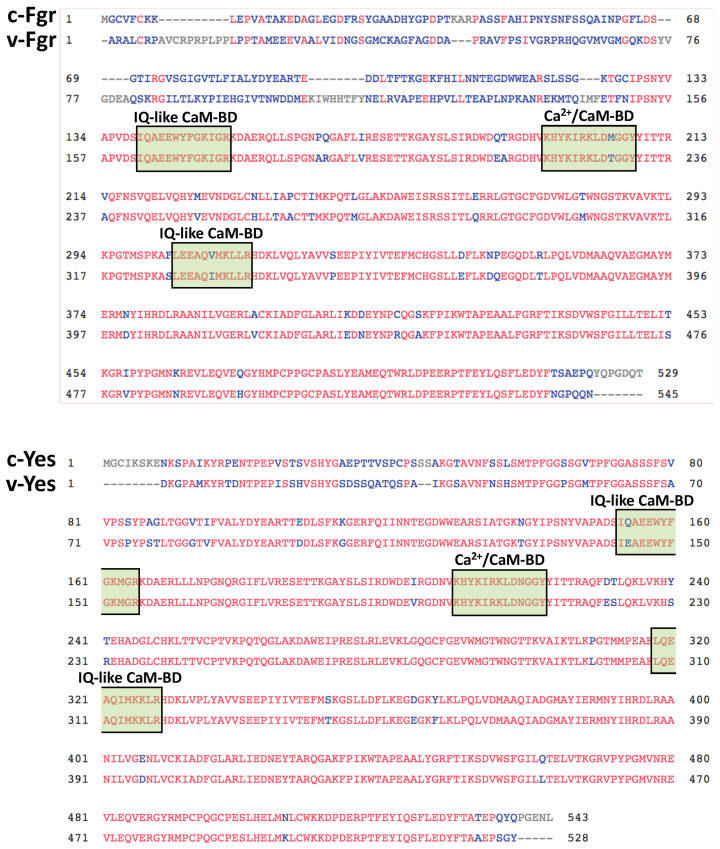
Comparison of the amino acid sequences of human c-Fgr and c-Yes with their viral counterparts. The plot shows the alignment of the amino acid sequences of human c-Fgr (P09769.2) [27] with Gardner–Rasheed feline sarcoma virus v-Fgr (P00544.1) [28] and human c-Yes (P07947) [29] with avian sarcoma virus Y73 v-Yes (P00527.2) [30], as prepared with the Cobalt program from the NCBI. Identical residues are marked in red and variable residues in blue. Extra residues in either protein are marked in gray. The sequences of c-Fgr/v-Fgr and c-Yes/v-Yes, which are homologous to the putative IQ-like calmodulin-binding domains (IQ-like CaM-BD) and Ca^2+^/CaM-binding domain (Ca^2+^/CaM-BD) of c-Src/v-Src, are marked with green boxes.

**Figure 3 biomolecules-13-01739-f003:**
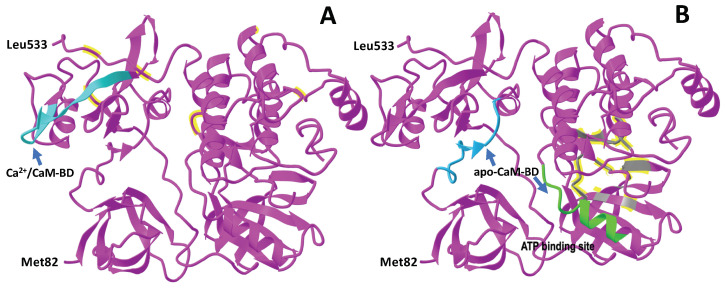
The CaM binding sites in c-Src and v-Src. (**A**,**B**) The figure depicts the crystallographic structure of a large fragment of human c-Src (from Met82 to Leu533), as described in [6], and taken from MCBI (1FMK), showing the proposed CaM-binding sites. The proposed Ca^2+^/CaM-binding site with sequence ^204^KH**Y**K**I**RK**L**DSGG**F**^216^ (residues 204–214 correspond to the GST-tagged protein), marked in cyan, located in the SH2 domain, was described as a basic 1-5-10 CaM-binding motif according to [37]. Two atypical IQ-like motifs were proposed as putative apo-CaM binding sites, as described in [43]. The ^143^**IQ**AEEWYFG**K**IT**R**^155^ sequence, located in the proximal region of the SH2 domain, is marked in blue, and the ^308^**LQ**EAQVM**KK**L**R**^318^ sequence, located in the proximal region of the tyrosine kinase domain, is marked in green. The ATP-binding site is marked in gray. The position numbers of the amino acids indicated in the marked sequences in this figure, as well as in the text, are directly determined from the crystallographic structure shown and are three positions lower from those mentioned in references [37,43]. (**C**) The plot shows the alignment of the amino acid sequences of human c-Src (NCBI ID: NP_938033.1) [53] and the Rous sarcoma virus v-Src (NCBI ID: P00524.6) [54], as prepared with the Cobalt program from the NCBI. Identical residues are marked in red, variable residues in blue, and the 3 extra residues at position 29–31, plus 7 extra residues in the C-terminus of c-Src, absent in v-Src, are marked in gray. The boxes highlighted in green indicate the Ca^2+^-dependent CaM-binding domain (Ca^2+^/CaM-BD) [37] and the two proposed atypical IQ-like CaM-BDs [43], likely representing apo-CaM binding sites.

**Figure 4 biomolecules-13-01739-f004:**
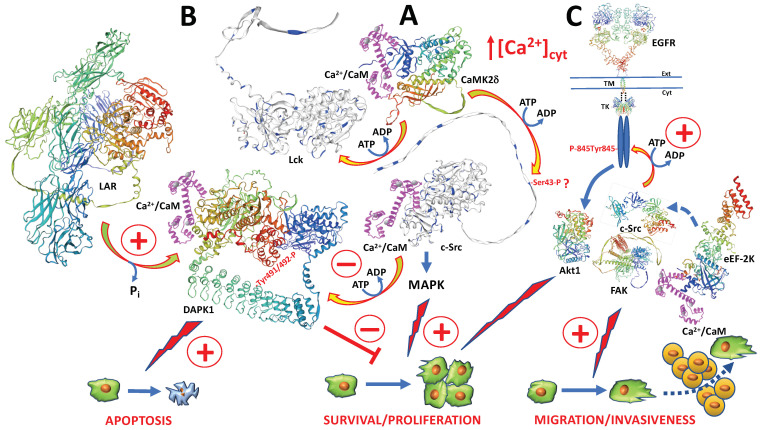
Action of CaM-dependent Ser/Thr-kinases on SFKs**.** (**A**) Upon increasing the cytosolic Ca^2+^ concentration ([Ca^2+^]_cyt_) and formation of the Ca^2+^/CaM complex, CaMK-II is activated and phosphorylates c-Src and Lck. In the case of c-Src, Ser43 could be a potential phosphorylation site. Activation of the CaMK-II/c-Src/MAPK pathway induces cell proliferation. (**B**) The Ca^2+^/CaM-dependent DAPK1 induces apoptosis, its phosphorylation at Tyr491/492 by c-Src causes its inactivation, and the receptor tyrosine-phosphatase LAR dephosphorylates phospho-Tyr491/492 reactivating DAPK1 enhancing apoptosis, inhibiting cell survival and cell proliferation. (**C**) The Ca^2+^/CaM-dependent eEF-2K controls c-Src, which phosphorylates and activates the EGFR at Tyr845, activating the Akt1-mediated cell survival pathway and c-Src/FAK-mediated tumor cell migration and invasiveness. For more details, see text. The structures of mouse Akt1 (P31750) [76], human CaM (1CLL) [12], mouse CaMK2δ (Q6PHZ2) [77], human c-Src (P12931) [78], human DAPK1 (AAI43760) [79], human eEF-2K (O00418) [80], human EGFR (P00533) [81], human FAK (Q05397) [82], human Lck (P06239) [58], and human LAR (P10586) [83], mostly depicted in a rainbow color code going from the N-terminus (blue) to the C-terminus (red), were built using the UniProt SWISS-MODEL database from Biozentrum, University of Basel, feeding the system with the amino acid sequences obtained from the NCBI (National Center for Biotechnology Information). Akt1, protein kinase B1; CaM, calmodulin; CaMK2δ, CaM-dependent protein kinase-2; c-Src, cellular sarcoma kinase homologue; DAPK1, death-associated protein kinase-1; eEF-2K, eukaryotic elongation factor-2 kinase; EGFR, epidermal growth factor receptor; FAK, focal adhesion kinase; Lck, lymphocyte-specific protein tyrosine kinase; LAR, leukocyte common antigen-related.

**Table 1 biomolecules-13-01739-t001:** Comparison of the sequences of the CaM-binding sites of human c-Src with homologous sequences in other human Src-family kinases.

SFK	Ca^2+^/CaM-BD in Distal SH2 Domain [37]	IQ-like apo-CaM-BD in Proximal SH2 Domain [43]	IQ-like apo-CaM-BD in Proximal TK Domain [43]
c-Src	^203^KHYKIRKLDSGGF^215^	^146^IQAEEWYFGKITR^158^	^311^LQEAQVMKKLR^321^
Yes	^210^KHYKIRKLD**N**GG**Y**^222^	^153^IQAEEWYFGK**MG**R^165^	^318^LQEAQ**I**MKKLR^328^
Fyn	^201^KHYKIRKLD**N**GG**Y**^213^	^144^IQAEEWYFGK**LG**R^156^	^312^L**E**EAQ**I**MKKL**K**^322^
Fgr	^196^KHYKIRKLD**M**GG**Y**^208^	^139^IQAEEWYFGKI**G**R^151^	^304^L**E**EAQVMK**L**LR^314^
Lyn	^181^KHYKIR**S**LD**N**GG**Y**^193^	^124^**LET**EEW**F**F**KD**ITR^136^	^288^L**E**EA**NL**MK**T**L**Q**^298^
Hck	^196^KHYKIR**T**LD**N**GGF^208^	^139^**LET**EEW**F**F**KG**I**S**R^151^	^303^L**A**EA**N**VMK**T**L**Q**^313^
Lck	^180^KHYKIR**N**LD**N**GGF^192^	^122^**LEP**E**P**W**F**F**KNLS**R^134^	^286^L**A**EA**NL**MK**Q**L**Q**^296^
Blk	^175^KHYKIR**C**LD**E**GG**Y**^187^	^119^**LEM**E**R**W**F**F**RSGG**R^131^	^282^L**G**EA**N**VMK**A**L**Q**^292^
Frk	^163^KHY**R**I**KR**LD**E**GGF^175^	^111^**L**QAE**P**W**F**FG**A**I**G**R^123^	^275^L**R**EAQ**I**MK**N**LR^285^

**Table 2 biomolecules-13-01739-t002:** SFK/Ca^2+^ signaling processes of oncological interest.

SFK	Tumor Type (Cells)	SFK and Ca^2+^ Signaling Findings	Refs.
c-Src, v-Src, Lck, Fyn	Assays in vitro with recombinant SFKs. Useful in many SFK expressing tumors	N^4^-derivatized and C(3)-derivatized analogs of PP1 present increased capacity to inhibit SFKs, and therefore SFK-mediated Ca^2+^ mobilization	[139]
c-Src	Pancreatic carcinoma (MiaPaCa-2, BxPC-3, AsPC-1 cells)	Blocking the Ca^2+^/CaM-BD of c-Src inhibits cell proliferation	[37,38]
c-Src	Pancreatic carcinoma (PANC-1 cells)	CaM antagonists (TFP and TMX) enhances apoptosis in tumor cells resistant to treatment with the anti-death receptor-5 antibody TRA-8 by preventing CaM/Src interaction	[140]
c-Src	Triple-negative breast carcinoma (MDA-MB-436 cells)	miRNA-603 blocks expression of CaM-regulated eEF-2K, and hence inhibits cell proliferation, invasiveness, tumor-associated angiogenesis and progression mediated by c-Src and other effectors	[141]
c-Src	Bcr-Abl positive chronic myeloid leukemia (K562 cells)	Combined inhibition of c-Src and PI_3_K in Imatinib-resistant Bcr-Abl positive CML cells induces disruption of Ca^2+^ mobilization, apoptosis, and autophagy	[142]
c-Src	Bcr-Abl positive chronic myeloid leukemia	The dual c-Src/Bcr-Abl inhibitor bosutinib also inhibits CaMK-IIγ	[143]
c-Src	Different cancers	Doxorubicin treatment induces heart and ovarian toxicity disrupting c-Src-mediated Ca^2+^ mobilization systems	[144,145]
c-Src	Schwannoma (patients isolated MPNST cells)	Combined inhibition of c-Src with sacaratinib and CaM with TFP decrease proliferation and cell survival	[146]
SFKs (ND)	Burkitt’s lymphoma (Ramos B cells)	Cholesterol depletion from lipid rafts inhibits SFK-dependent Ca^2+^ mobilization and apoptosis induced by the anti-CD20 antibody rituximab	[147,148]
Lyn	Different cancers; e.g., genito-urinary tumors (macrophages isolated from BALB/c mice)	Cisplatin treatment induces expression and activation of Lyn, and inhibitors of Ca^2+^ mobilization and CaM antagonists inhibit Lyn activation	[149]
Lyn	Mastocytoma (P815 cells)	Allergen-induced degranulation of mast cells induced Lyn activation and Ca^2+^ released by the IP_3_ receptor	[150]
Fyn, Lck	DMBA-induced T cells leukemia (HPB-ALL cells)	DMBA activates Fyn and Lck inducing PLCγ activation, IP_3_ production and Ca^2+^ released from the endoplasmic reticulum	[151]
Lck	Thymus-derived mouse lymphoma (WEHI 7.1 cells)	Dexamethasone-induced inhibition of Lck changes the pattern of TCR-mediated Ca^2+^ signals from transient to oscillatory by downregulating IP_3_ receptors	[152]
Fyn	Astrocytoma (1321N1 cells)	Ca^2+^ mobilization induced by 5-HT_6_ activation induces Fyn activation	[153]

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
