# Peer review of "Ca2+ Signaling and Src Functions in Tumor Cells"

_biomolecules, 2023, doi:10.3390/biom13121739_

Round 1
Reviewer 1 Report
Comments and Suggestions for Authors
This is an interesting review summarizing current knowledge on the link between calcium homeostasis and Src signalling. In this review a very large number of studies list Ca-related events affecting Src-dependent biological activities, which makes it sometimes difficult for the reader to follow. This manuscript would benefit from figures and tables that would list Ca-related events and outcomes/mechanisms related to Src activity.
1. The current figure 1 and 3 provide structures of Src and other Src family members. In relation to this review, it would be more helpful to provide schemes/drawings that indicate the various domains and how they might be affected by Ca-signalling.
2. The current Figure 2 provides the full sequence of c- and v-Src. Rather than providing the full sequence, just highlighting the CaM binding sites and differences at the C-terminus would be sufficient. This could be combined/incorporated with current Figure 5.
3. Figure 4 shows the sequence of c-Fgr and cYes and viral counterparts. Unless this figure highlights motifs or amino acids that are relevant in relation to calcium, this should be removed.
4. Figure 6: rather than showing the whole AA sequence, just highlighting the CaM domains would be sufficient.
5. The author should consider to provide tables that list Ca-dependent events and Src-related outcome. References for these points should also be provided in these tables. These tables should provide an overview of chapter 3-4 (direct interaction of CaM with Src; CaM phosphorylation by Src), Chapter 5-7 (CaM-dependent kinases and Src; Calcinuerin and SFKs; Ca and Src degradation), chapter 8-9 (Ca, Src and cancer) and chapter 10 (Medical implications)
6. The author should also consider a graph (possibly with a few panels) that summarize/visualize how Calcium may regulate Src.
Comments on the Quality of English LanguageOverall, the review would benefit from minor editing.
Author Response
This is an interesting review summarizing current knowledge on the link between calcium homeostasis and Src signalling. In this review a very large number of studies list Ca-related events affecting Src-dependent biological activities, which makes it sometimes difficult for the reader to follow. This manuscript would benefit from figures and tables that would list Ca-related events and outcomes/mechanisms related to Src activity.
Thank you for the good reception of the review and for providing suggestions to improve the manuscript. As suggested, we improved figures and added tables listing Ca2+-related events and Src functions. In what follows, it is described point-by-point the answer to specific comments and the changes done in the revised version.
- The current figure 1 and 3 provide structures of Src and other Src family members. In relation to this review, it would be more helpful to provide schemes/drawings that indicate the various domains and how they might be affected by Ca-signalling.
Figure 1 has been extensively modified following the suggestion of the reviewer. We added in the new Figure 1 panel B, drawings of the different domains of c-Src illustrating its inactive and active conformations, the phosphorylation of calmodulin (CaM) by c-Src and the interaction of phospho-CaM with the kinase, as well as c-Src-mediated Ca2+ mobilization, and its role in forming the Ca2+/CaM complex and its binding to the kinase. Nevertheless, the crystal structure of c-Src is maintained in panel A, as it is important that readers obtain a more realistic view of the kinase. In contrast, we have deleted Figure 3, as it does not add useful information to the topic of this review, and the homologous sequences of the CaM binding sites of c-Src in the different SFKs are now shown in Table 1.
- The current Figure 2 provides the full sequence of c- and v-Src. Rather than providing the full sequence, just highlighting the CaM binding sites and differences at the C-terminus would be sufficient. This could be combined/incorporated with current Figure 5.
As suggested, we have incorporated old Figure 2 to former Figure 5 (new Figure 3). However, the full sequences of c-Src and v-Src remain, as it is extremely difficult to document only the three CaM-binding sites and the C-terminal region of the proteins in a single panel, as they are widely separated. Also, in this manner, changes in other regions in the sequences in c-Src and v-Src are also documented.
- Figure 4 shows the sequence of c-Fgr and cYes and viral counterparts. Unless this figure highlights motifs or amino acids that are relevant in relation to calcium, this should be removed.
As requested, we have highlighted in old Figure 4 (now new Figure 2) the sequences of c-Fgr/v-Fgr and c-Yes/v-Yes corresponding to the homologous sequences of the three CaM-binding sites in c-Src/v-Src.
- Figure 6: rather than showing the whole AA sequence, just highlighting the CaM domains would be sufficient.
Old Figure 6 has been substituted with Table 1, containing the sequences in the different human SFKs that are homologous to the sequences of the three CaM binding sites of human c-Src, as requested.
- The author should consider to provide tables that list Ca-dependent events and Src-related outcome. References for these points should also be provided in these tables. These tables should provide an overview of chapter 3-4 (direct interaction of CaM with Src; CaM phosphorylation by Src), Chapter 5-7 (CaM-dependent kinases and Src; Calcinuerin and SFKs; Ca and Src degradation), chapter 8-9 (Ca, Src and cancer) and chapter 10 (Medical implications)
We have added Table 2 listing, with the corresponding references, the major Ca2+/Src signaling events of interest in the field of Oncology. However, we have not added tables to the rest of the sections, as we consider un unnecessary redundant with the information that is provide in great detail in the text. Instead, we have added sub-titles to the different sections to better understand their contents, as requested by another reviewer.
- The author should also consider a graph (possibly with a few panels) that summarize/visualize how Calcium may regulate Src.
In new Figure 1 panel B, we included the phosphorylation of calmodulin (CaM) by c-Src and the interaction of phospho-CaM with the kinase, as well as c-Src-mediated Ca2+ mobilization, and its role in forming the Ca2+/CaM complex and binding to the kinase. I believe that this is sufficient, as it illustrates the major points of how Ca2+ regulates c-Src.
Reviewer 2 Report
Comments and Suggestions for Authors
This paper reviews the role of Ca2+ signaling and proto-oncogene non-receptor tyrosine kinase c-Src in tumor cells. In particular, the interaction between CaM and Src, the phosphorylation of CaM by Src, and the role of CAM-dependent phosphatase calcineurin in v-Src/c-Src are discussed. This paper also prospected the therapeutic effect of calcium signal-regulated Src in human tumors.
Comments
1. The content of the paper covers a large amount of words, so it is recommended to set sub-titles so that readers can understand the main content of the paper relatively quickly.
2. The structure of Src is introduced in the introduction section, and it is suggested to use a cartoon illustration to describe the content.
3. The pictures of molecular docking are perfunctory. It is suggested that the functional domain of the molecule is best illustrated directly in the diagram. For example, the blue band in Figure 5 represents CaM binding sites. Please mark the location of the blue ribbon as "CaM binding site". What exactly do the six graphs in Figure 3 mean? Where are the similarities and differences between these six molecules? Please mark them appropriately in the figure.
4. As my understanding, Figure 7 is an unverified result. This part of content can be written in a future research article, and it is suggested to delete it in this paper.
5. The clarity and elegance of the images of sequence alignment need to be improved.
Author Response
This paper reviews the role of Ca2+ signaling and proto-oncogene non-receptor tyrosine kinase c-Src in tumor cells. In particular, the interaction between CaM and Src, the phosphorylation of CaM by Src, and the role of CAM-dependent phosphatase calcineurin in v-Src/c-Src are discussed. This paper also prospected the therapeutic effect of calcium signal-regulated Src in human tumors.
Thank you for the good reception of this work and for providing helpful suggestions to improve the manuscript.
Comments
- The content of the paper covers a large amount of words, so it is recommended to set sub-titles so that readers can understand the main content of the paper relatively quickly.
We have added subtitles in some relevant sections as requested.
- The structure of Src is introduced in the introduction section, and it is suggested to use a cartoon illustration to describe the content.
As requested, a cartoon describing details the structure of c-Src, together with other items requested by the other reviewer, has been introduced in new Figure 1.
- The pictures of molecular docking are perfunctory. It is suggested that the functional domain of the molecule is best illustrated directly in the diagram. For example, the blue band in Figure 5 represents CaM binding sites. Please mark the location of the blue ribbon as "CaM binding site". What exactly do the six graphs in Figure 3 mean? Where are the similarities and differences between these six molecules? Please mark them appropriately in the figure.
We have marked in former Figure 5 (new Figure 3) the three CaM-binding sites, but for clarity, consistence with the text, and limited space to include the full name in the figure, we used Ca2+/CaM-BD to mark the site of Ca2+/calmodulin binding, and apo-CaM-BD to mark the site of Ca2+-free calmodulin binding, where BD means binding domain. We have deleted old Figure 3, as requested by another reviewer, and as we now believe that really it does not add useful information to the topic of this review. The sequences in the different human SFKs that are homologous to the sequences of the three CaM-binding sites of human c-Src are now shown in Table 1.
- As my understanding, Figure 7 is an unverified result. This part of content can be written in a future research article, and it is suggested to delete it in this paper.
Old Figure 7 has been deleted as requested. Also, Ser43 in human c-Src within the sequence 40KPASAD45 coincides with the phosphorylation motif of CaMK-IId of general sequence -(R/K)-X-X-p(S/T)-X-(D/E), where X means any amino acid. This information has been added together with a couple of relevant references about the preferred phosphorylation motif of CaMK-IId in target substrates.
- The clarity and elegance of the images of sequence alignment need to be improved.
The alignment of the full sequences of the Src family kinases has been deleted and substituted by Table 1. The alignment of c-Src and v-Src (new Fig. 3C) has been improved.
Round 2
Reviewer 1 Report
Comments and Suggestions for Authors
The authors have addressed all points raised by this reviewer.
Comments on the Quality of English LanguageMinor editing of English language required.
Reviewer 2 Report
Comments and Suggestions for Authors
The revise is fine, except for the position of "A", "B", and so on in the Figures. The author needs to unify the position of the ABC marks in the pictures, whether it is on the top left, top right, or middle, to improve readability.